# Targeted High-Throughput Sequencing Enables the Detection of Single Nucleotide Variations in CRISPR/Cas9 Gene-Edited Organisms

**DOI:** 10.3390/foods12030455

**Published:** 2023-01-18

**Authors:** Marie-Alice Fraiture, Jolien D’aes, Emmanuel Guiderdoni, Anne-Cécile Meunier, Thomas Delcourt, Stefan Hoffman, Els Vandermassen, Sigrid C. J. De Keersmaecker, Kevin Vanneste, Nancy H. C. Roosens

**Affiliations:** 1Sciensano, Transversal Activities in Applied Genomics (TAG), rue Juliette Wytsman 14, 1050 Brussels, Belgium; 2CIRAD, UMR AGAP Institut, F-34398 Montpellier, France; 3UMR AGAP Institut, Université de Montpellier, CIRAD, INRAE, Institut Agro, F-34398 Montpellier, France

**Keywords:** CRISPR/Cas9, single nucleotide variations, transgenic rice, detection, PCR-based enrichment, high-throughput sequencing, food and feed chain, genetically modified organism

## Abstract

Similar to genetically modified organisms (GMOs) produced by classical genetic engineering, gene-edited (GE) organisms and their derived food/feed products commercialized on the European Union market fall within the scope of European Union Directive 2001/18/EC. Consequently, their control in the food/feed chain by GMO enforcement laboratories is required by the competent authorities to guarantee food/feed safety and traceability (2003/1829/EC; 2003/1830/EC). However, their detection is potentially challenging at both the analytical and interpretation levels since this requires methodological approaches that can target and detect a specific single nucleotide variation (SNV) introduced into a GE organism. In this study, we propose a targeted high-throughput sequencing approach, including (i) a prior PCR-based enrichment step to amplify regions of interest, (ii) a sequencing step, and (iii) a data analysis methodology to identify SNVs of interest. To investigate if the performance of this targeted high-throughput sequencing approach is compatible with the performance criteria used in the GMO detection field, several samples containing different percentages of a GE rice line carrying a single adenosine insertion in *OsMADS26* were prepared and analyzed. The SNV of interest in samples containing the GE rice line could successfully be detected, both at high and low percentages. No impact related to food processing or to the presence of other crop species was observed. The present proof-of-concept study has allowed us to deliver the first experimental-based evidence indicating that the proposed targeted high-throughput sequencing approach may constitute, in the future, a specific and sensitive tool to support the safety and traceability of the food/feed chain regarding GE plants carrying SNVs.

## 1. Introduction

Gene-editing techniques such as CRISPR/Cas9 have revolutionized the genetic manipulations of organisms, allowing the introduction of a precise modification (substitution, insertion, or deletion) in the genome of an organism of interest. Such modifications include both large nucleotide sequence variations as well as single nucleotide variations (SNVs) [1,2,3,4,5,6,7]. Given the endless possibilities offered by these techniques, the commercialization of such gene-edited (GE) organisms by biotech companies is expected to increase in the next few years. Currently, a GE high-GABA (gamma-aminobutyric acid) tomato has been commercialized in Japan, while two GE crops, a herbicide-tolerant canola and a soybean with a modified oil composition, have been commercialized in the United States of America [1,2,3,4,5,6,7].

Since 2018, the commercialization of GE organisms and their derived food/feed products on the European Union market has fallen within the scope of European Union Directive 2001/18/EC, similar to genetically modified organisms (GMOs) produced by classical genetic engineering. To guarantee the traceability of such products in the food/feed chain, control by the competent authorities using enforcement laboratories is therefore requested [2,4,7,8,9,10,11,12]. For GE organisms carrying a large nucleotide sequence variation, the development of detection methods is highly similar to the detection of GMOs produced by classical genetic engineering. However, for GE organisms carrying an SNV of only one or a few nucleotides, their detection and discrimination from the parental lines are not trivial, both at the analytical and interpretation levels. At the analytical level, the design of a detection method restricted to a particular region carrying such an SNV of interest can be challenging. At the interpretation level, successful detection of an SNV does not automatically prove the use of gene-editing techniques because an SNV of only one or few nucleotides could also have occurred naturally or through random mutagenesis techniques [2,4,7,12]. 

Among the available detection methods, the potential use of PCR-based methods, such as real-time PCR and digital PCR technologies that are widely used by GMO enforcement laboratories for the detection of GMOs produced by classical genetic engineering, has already previously explored and has been assessed as suitable to analytically detect crops carrying an SNV introduced by gene editing [3,13,14,15,16]. However, without information demonstrating that the SNV is only present in a particular GE organism, the presence of this GE organism cannot be unambiguously deduced from a positive PCR signal associated with this SNV [2,3,13,14,15,16,17]. Therefore, in addition to detecting the SNV(s) introduced by gene-editing techniques, the detection of additional targets is required to identify the specific line that was GE. These targets can include, for example, certain nucleotide variations associated specifically with a particular genotype or cultivar as well as potential off-target mutations introduced through the gene-editing process. Unintended changes to the genome can indeed occur using gene-editing techniques, such as CRISPR/Cas9, at sites with high sequence similarity to the on-target sequence, although gene-editing processes are generally designed to minimize them and their frequency is usually low [2,18,19]. However, the design and development of real-time PCR and digital PCR methods to simultaneously detect multiple targets of interest can quickly become fastidious, as multiplexing with these technologies is usually limited to only a few targets [20,21,22]. Therefore, such PCR-based technologies, although widely used in GMO enforcement laboratories for the detection of classical GMOs, do not necessarily represent the most appropriate strategy in this context. 

A targeted high-throughput sequencing strategy can overcome these bottlenecks as it offers both sequence information and a high potential for multiplexing by simultaneously targeting multiple samples as well as several SNVs of interest [2,18,23,24,25]. This sequencing strategy requires a prior enrichment step of the principal targets using probe-based capture hybridization or PCR-based amplification. The latter enrichment approach is usually considered the most efficient and presents the advantage of avoiding potential difficulties related to the design of probes specific to SNVs. Targeted high-throughput sequencing strategies have previously been applied in different fields, including the genotyping of plants based on key SNVs [2,18,23,24,25]. The potential ability of high-throughput sequencing, including targeted strategies, to be used for the detection of GMOs produced by classical genetic engineering has also been previously illustrated [26,27,28,29,30,31,32,33,34,35,36,37,38,39]. However, in contrast to PCR-based methods (e.g., conventional PCR, real-time PCR, digital droplet PCR (ddPCR)), high-throughput sequencing-based methods still require further harmonization and standardization to be used in the context of GMO control. Nowadays, in the GMO detection field, no minimum quality performance criteria are established for sequencing-based methods [40]. In addition, to our knowledge, no targeted high-throughput sequencing strategy has currently yet been experimentally investigated for the detection of SNVs carried by GE organisms, leading to a lack of information about the performance of the methodology and the potential compatibility of such methodology with the European GMO regulations, including the threshold of 0.9% for the labeling of authorized GMOs and the technical zero-tolerance threshold of 0.1% for the low-level presence of GMOs in feed. In the case of non-European countries, for example, in Japan, a threshold of 5% is established for GMO labeling [9,41,42].

In this proof-of-concept study, for the first time, a targeted high-throughput sequencing strategy has been developed to detect, in multiple samples in parallel, an SNV of interest carried by a GE plant, and the performance of this approach is assessed. A GE rice line carrying a single adenosine insertion in *OsMADS26* was used. *OsMADS26*, encoding a MADS-box transcription factor, was previously reported as putatively involved in rice stress responses [16]. Several samples mimicking various contaminations of this GE rice line in food products were prepared to assess the feasibility and performance of the proposed targeted high-throughput sequencing approach (e.g., specificity, sensitivity, applicability). The generated sequencing data were analyzed using an in-house pipeline developed to identify SNVs, including both single polymorphism nucleotides and indels, as well as to estimate their frequency. The potential added value of the proposed targeted high-throughput sequencing to support GMO enforcement laboratories in the control of GE organisms in the food/feed chain is discussed in light of the experimental-based results obtained in this proof-of-concept study.

## 2. Materials and Methods

### 2.1. Plant Materials

Rice (*Oryza sativa* L. *Nipponbare*) seeds from a GE line and its parental line were used in this study. The GE rice line, carrying a single adenosine insertion in *OsMADS26* (locus: Os08g02070) encoding a MADS-box transcription factor, was generated using CRISPR/Cas9 and is currently not commercialized on the market. The homozygous adenosine insertion was integrated into the coding region close to the start codon between genomic positions 679646 and 679647 on chromosome VIII. This frameshift mutation inactivates *OsMADS26*, consequently putatively increasing biotic resistance and biotic stress tolerance (Rice SNP-Seek Database; Meunier et al., unpublished) [16,43]. Using rice seeds from the GE line and its parental line, rice noodles were prepared in-house as described previously [16]. DNA from homogenous powders of ground rice seeds and rice noodles was extracted using a CTAB-based procedure [44] in combination with the Genomic-tip20/G kit (QIAGEN, Hilden, Germany), as described previously [45,46]. 

Certified reference materials of wild-type (WT) maize (*Zea mays*) and WT soybean (*Glycine max*) were collected from the Institute for Reference Materials and Measurements (IRMM, Geel, Belgium). DNA from these crop species was obtained as described previously [47]. 

Using the Nanodrop^®^ 2000 (ThermoFisher, Wilmington, DE, USA) device, the DNA concentration was measured by spectrophotometry. DNA purity was considered as complying with the acceptance criteria (A260/A280 ration ~1.8 and A260/A230 ratio ~2.0–2.2).

### 2.2. Sample Preparation

To assess the performance of the proposed targeted high-throughput sequencing approach, several samples were prepared. For samples n°1–11 (Table 1), DNA from rice seeds of the GE rice line and its parental line were mixed to create samples containing 100%, 99.9%, 99.1%, 95%, 90%, 50%, 10%, 5%, 0.9%, 0.1%, or 0% of the GE rice line, ranging from ~14,000 to 0 estimated haploid genome copies [16] (Appendix A). 

For samples n°12–17 (Table 2), DNA from in-house-prepared rice noodles of the GE rice line and its parental line were mixed to obtain samples containing 100%, 99.9%, 99.1%, 0.9%, 0.1%, or 0% of the GE rice line, ranging from ~14,000 to 0 estimated haploid genome copies [16] (Appendix A). 

For samples n°18–19 (Table 3), DNA from rice seeds of the GE rice line and its parental line, both at 14 estimated haploid genome copies, were mixed with DNA from WT maize or WT soybean at 13,972 estimated haploid genome copies [16] (Appendix A). 

The calculation of the estimated haploid genome copy number was based on the size of the rice (0.5 pg), maize (2.6 pg), and soybean (1.16 pg) haploid genomes [48,49].

### 2.3. ddPCR Assays

A recently developed 2-plex ddPCR method targeting the GE rice line was applied in triplicate to all samples (n°1–19), as described previously [16]. For each ddPCR assay, an NTC (no template control) was included. On this basis, the copy number of the GE rice line was measured (Appendix A).

### 2.4. Conventional PCR Assays

A standard 25 µL reaction volume was used, containing 1x KAPA HiFi HotStart ReadyMix (Roche, Brussels, Belgium), 900 nM of each primer (Eurogentec, Liège, Belgium), and 5 µL of DNA. The primer pair (OsMADS26-F: GACAGGAGGAGAGGAGGAAGA; OsMADS26-R: AAGGTGACCTGACGGTGAAC) used in this study was previously designed to cover 113 bp, for the GE rice line, and 112 bp, for the parental rice line, of the *OsMADS26* region carrying the CRISPR/Cas9 SNV (Fraiture et al., 2022). In this study, these primers were supplemented by Illumina^®^ overhang adapter sequences (OsMADS26-F: TCGTCGGCAGCGTCAGATGTGTATAAGAGACAG-[GACAGGAGGAGAGGAGGAAGA]; OsMADS26-R: GTCTCGTGGGCTCGGAGATGTGTATAAGAGACAG-[AAGGTGACCTGACGGTGAAC]) in order to be compatible with the subsequent Illumina^®^ MiSeq sequencing run (Illumina support instructions). The PCR assays were performed on a T100TM Thermal Cycler (Bio-Rad, Temse, Belgium). The PCR program consisted of a single cycle at 95 °C for 3 min (initial denaturation), 30 cycles at 95 °C for 30 s (denaturation), at 60 °C for 30 s (annealing), and at 72 °C for 30 s (extension), and a single cycle at 72 °C for 5 min (final extension). The final PCR products were visualized by electrophoresis using the Tapestation 4200 device with the associated D1000 Screen Tape and reagents (Agilent, Machelen, Belgium) (Appendix A). Each sample (n°1–19) and the included NTC (no template control) were tested in quadruplicate.

### 2.5. Library Preparation, Sequencing, and Data Analysis

Following the manufacturer’s instructions, each PCR product was purified using Agencourt^®^ Ampure^®^ XP (Beckman Coulter, Danvers, MA, USA), and amplicon sequencing libraries were prepared according to Illumina’s instructions (Illumina, San Diego, CA, USA) (Illumina support instructions). Then, 15% PhiX was added to the library pool. Each of the 80 PCR products, generated from the 20 different samples (n°1–20), was individually barcoded and then pooled for sequencing on the same run. Sequencing was carried out on an Illumina^®^ MiSeq system using V3 chemistry, obtaining 250 bp paired-end reads.

The quality of raw sequencing data was evaluated using FastQC 0.11.5 (www.bioinformatics.babraham.ac.uk/projects/fastqc, accessed on 14 August 2021) with default settings (Appendix A). The frequency of the GE rice line in the samples was estimated with an in-house pipeline, consisting of the alignment of the amplicon sequencing reads to a rice reference genome, followed by variant calling to detect variants and estimate their allelic frequency. The reference for the alignment, the reference genome of the *Oryza sativa* Japonica Group (RefSeq GCF_001433935.1), was indexed with Samtools 1.9 [50] and Bowtie2 2.3.4.3 [51], while Picard 2.8.14 was used to generate a dictionary of the indexed reference FASTA file. The raw sequencing data were pre-processed using Trimmomatic 0.38 [52] with the following settings: ILLUMINACLIP:NexteraPE-PE.fa:2:30:10, LEADING:10, TRAILING:10, SLIDINGWINDOW:4:20, MINLEN: 40, after which the trimmed reads were aligned to the reference with Bowtie2 with the “--end-to-end” and “--very-sensitive” settings. The resulting alignments in SAM format were converted to BAM format with Samtools and sorted using Picard [53] with the option “SORT_ORDER = coordinate”, followed by assignment of the reads to a new read group with Picard, with the flags “LB”, “PL”, “PU”, and “SM” set to the arbitrary placeholder value “test.” The resulting BAM files were indexed using Samtools and used as input for the creation of a target intervals file and indel realignment using GATK 3.7 [54], with “–maxReads” set to 1,000,000. Samtools was used to index the generated BAM files, followed by the insertion of indel qualities into the BAM files with LoFreq 2.1.3.1 [55]. Next, variants were called with the LoFreq package using the options “--call-indels”, “--min-bq 35”, “--min-mq 42”, and “--sig 0.001”. Additionally, the output calls were limited to positions 679,589–679,659 of chromosome 8 of the reference (NC_029263.1), which represents the amplicon sequence targeted by the PCR assay described in Section 2.3, excluding the primer sequences. No off-target mutation was observed in the target amplicons. The resulting VCF file was filtered with LoFreq, setting the minimal allelic frequency to 0.01 %. All samples were analyzed in this study with a cut-off for the allelic frequency at 0.1%.

## 3. Results

### 3.1. Development of a Workflow for Targeted High-Throughput Sequencing

The possibility of using a targeted high-throughput sequencing approach for the detection of an SNV in CRISPR/Cas9 plants has been investigated in this study. As a proof-of-concept, a GE rice line carrying a homozygous single adenosine insertion in *OsMADS26* and its parental rice line were used. Using CRISPR/Cas9, this SNV was introduced in the coding region close to the start codon, resulting in a gene inactivation that was expected to increase the biotic resistance and biotic stress tolerance of the GE rice line (Meunier et al., unpublished) [16,56]. In this study, a targeted high-throughput sequencing approach has been investigated for the detection of the single nucleotide insertion carried by the GE rice line. This sequencing approach is composed of three main successive steps (Figure 1). 

First, a PCR-based enrichment step is applied to the whole DNA extracted from a given food matrix. A primer pair (OsMADS26-F and OsMADS26-R), previously designed to amplify the *OsMADS26* region carrying the SNV of interest, was used (Fraiture et al., 2022) to create two different PCR amplicons: one with a length of 113 bp for the GE rice line and one with a length of 112 bp for the parental rice line (Figure 1). The specificity of this primer pair has been demonstrated previously [16]. Among the entire NCBI nucleotide (nr/nt) collection, only hits with sequences belonging to rice species were observed for these PCR amplicons [16]. Moreover, at the experimental level, PCR amplification was only observed with rice materials, both using WT and transgenic rice lines, and no PCR amplification was observed for non-targeted DNA, including various animal, microbial, and plant materials [16]. Second, each PCR product from each sample is individually coupled to a unique barcode during the sequencing library preparation steps, and all barcoded PCR products are subsequently pooled for high-throughput sequencing. The multiplexing of several samples, as illustrated in this study, can also be used to target additional SNVs of interest if these are known and/or available (e.g., species/genotype-specific sites and off-targeting signatures). Finally, based on the associated unique barcode, all raw sequencing data are demultiplexed. For each PCR product, the raw sequencing reads are analyzed using an in-house pipeline that compares each nucleotide from the generated amplicons to a rice reference genome, allowing us to (i) identify SNVs, including both single nucleotide polymorphisms and small indels, and (ii) estimate the frequency of the SNVs of interest. This pipeline was adapted from a previously developed pipeline to detect low-frequency variants in SARS-CoV-2 [57]. The pipeline input consists of the demultiplexed raw sequencing data, which are first processed to filter out low-quality reads and trim low-quality sequences from the reads. The resulting processed reads of the target amplicon are then aligned to the corresponding target region in the reference. To determine the frequency at which the SNV of interest is present in the aligned reads, variants are called for the target region with very strict quality cut-offs to retain only high-confidence variant calls and filtering to remove variants present at a frequency below 0.1%, reflecting the required threshold for the detection of GMOs in food/feed samples. 

In this study, the multiplexing and performance of the proposed targeted high-throughput sequencing approach to detect the CRISPR/Cas9 adenosine insertion found in the GE rice line are explored. To this end, as schematized in Figure 1, several samples were prepared, containing different percentages of the GE rice line and its parental line, and subsequently pooled to be processed in the same sequencing run.

### 3.2. Assessment of Sensitivity

To assess whether the proposed targeted high-throughput sequencing approach is able to detect GMO contaminations at trace levels, its sensitivity was investigated using samples n°1–11. For these mixture samples, DNA from rice seeds of the GE rice line and its parental line (DNA extraction yield: ~6 × 10^3^ ng/g of rice) were mixed to obtain samples containing 100%, 99.9%, 99.1%, 95%, 90%, 50%, 10%, 5%, 0.9%, 0.1%, or 0% of the GE rice line (Table 1). For each sample, the copy number associated with the GE rice line was measured by ddPCR (Table 1 and Appendix A). These mixture samples were specifically composed to mimic different contamination levels of a GE plant in its WT background. 

The presence of the single variation of interest was not detected in sample n°11, which was exclusively composed of the parental rice line, whereas this SNV was observed in all samples containing the GE rice line (n°1–10), both at high and low percentages (Table 1). This SNV was detected in samples containing as low as 0.9% for all four replicates of the GE rice line, corresponding to approximately 85 haploid genome copies. The SNV of interest was also observed for the majority of replicates in sample n°10, containing 0.1% of the GE rice line, corresponding to approximately 11 haploid genome copies. In addition, the observed frequency of the detected single variation of interest was close to the expected value for all tested samples. 

Based on these results, using the proposed targeted high-throughput sequencing approach, with an observed depth of coverage varying between 70,873 and 158,436 for the analyzed samples (n°1–11), the detection of the SNV of interest was possible even at low contamination levels (as low as 11 haploid genome copies and 0.1% of transgenic material per ingredient), which is crucial for GMO enforcement laboratories. The proposed targeted high-throughput sequencing approach was also able to deal with samples containing less than 25 copies of the target, which is one of the minimum performance criteria for GMO detection methods [40]. However, although the proposed targeted high-throughput sequencing approach is compatible with the sensitivity performance criteria used in the GMO detection field, such performance criteria are currently only intended for PCR-based detection methods (e.g., conventional PCR, real-time PCR, ddPCR), and no criteria have currently been established for sequencing-based detection methods. In addition, despite the initial objective of assessing the sensitivity of the proposed targeted high-throughput sequencing approach, the generated results also allowed us to provide preliminary data confirming the possible detection of heterozygous mutations. For example, sample n°6 may also be used to mimic a GE rice line carrying a heterozygous mutation (Table 1). 

### 3.3. Assessment of Applicability

The applicability of the proposed targeted high-throughput sequencing approach was investigated using two different types of samples. 

Firstly, samples n°12–17 were prepared to mimic different contamination levels of the GE rice line in its parental rice line within a processed food matrix, such as rice noodles. Food processing, defined as any physical, chemical, or mechanical food manipulations from the raw material to the final product, may induce DNA damage, such as the fragmentation of high molecular weight DNA. In order to mimic processed samples with strong DNA degradation, rice noodle materials were used in this study [58]. These rice noodle samples contained 100%, 99.9%, 99.1%, 0.9%, 0.1%, or 0% of the GE rice line (Table 2). For each sample, the copy number associated with the GE rice line was measured by ddPCR (Table 2 and Appendix A). The depth of coverage of the processed rice samples varied between 47,261 and 120,711. The SNV of interest was detected in all rice noodle samples containing the GE rice line, both at high and low percentages. The SNV was observed in all four replicates, with a frequency of the GE rice line as low as 0.1%, corresponding to approximately 14 haploid genome copies. Regarding the observed frequency of the SNV of interest, the values were close to the expected ones. In addition, the results obtained from the rice noodle samples (n°12–17) were generally similar to the results observed for the unprocessed rice seed samples (n°1–11), supporting that the proposed targeted high-throughput sequencing approach does not appear to be impacted by the DNA degradation of the analyzed samples.

Secondly, samples n°18–19 were prepared to mimic low-level contamination of both the GE rice line and its parental line in another crop species, such as maize or soybean (Table 3). More precisely, these samples contained 0.1% of the GE rice line (corresponding to 14 estimated haploid genome copies), 0.1% of the parental rice line (corresponding to 14 estimated haploid genome copies), and 99.8% of WT maize or WT soybean (corresponding to 13,972 estimated haploid genome copies). For each sample, the copy number associated with the GE rice line was measured by ddPCR (Table 3 and Appendix A). Although the observed depth of coverage of these crop mixture samples was lower than for the other analyzed samples, the presence of the SNV of interest was detected in the four replicates, with an estimated frequency close to the expected value. These results, therefore, similarly support that the proposed targeted high-throughput sequencing approach is not affected by the presence of a high percentage of untargeted materials from other plant species.

## 4. Discussion

Despite their increasing use to strengthen the safety and traceability of the food/feed chain at several levels [59], sequencing-based methods are currently not widely used by GMO enforcement laboratories. Further harmonization and standardization are indeed still required to be used in the context of GMO control. The proposed targeted high-throughput sequencing approach developed in this proof-of-concept study allowed us to generate data in order to contribute to the assessment of minimum quality performance criteria for the detection of GE plants using sequencing-based methods. To our knowledge, the present study is the first available study using a targeted high-throughput sequencing approach to detect SNVs of interest in a GE organism. As a proof-of-concept, a CRISPR/Cas9 GE rice line carrying a single adenosine insertion, currently not commercialized on the market, was used. Using several samples containing different percentages of the GE rice line, the feasibility and performance of this targeted high-throughput sequencing approach were investigated, for the first time, as a tool to enable GMO control, guaranteeing the safety and traceability of the food/feed chain. The targeted high-throughput sequencing approach was able to detect the SNV of interest, even at the low level of contamination by the GE rice line of 0.1%. Moreover, although more conditions will need to be tested, this targeted high-throughput sequencing approach does not seem to be affected by food processing and is compatible with a low contamination level of the GE rice line in another plant species. Furthermore, although this targeted high-throughput sequencing approach was used as a qualitative detection method, the observed frequency of the single nucleotide insertion of interest was generally close to the amount of the GE rice line contained in each tested sample. Consequently, the proposed targeted high-throughput sequencing approach was demonstrated to be a promising tool to detect specific SNVs in GE organisms as well as to estimate their frequency.

One of the main advantages of the proposed targeted high-throughput sequencing approach is its high-multiplexing capacity, as illustrated in this study through the simultaneous sequencing of several different samples. This high-multiplexing capacity could be used to detect, in parallel, additional variations introduced by different routes (e.g., natural occurrences, random mutagenesis, or gene editing) as well as potential off-targeting sites. Key targets could also be selected in order to identify the specific line used to introduce modifications through gene-editing techniques. Although it will not completely solve the bottlenecks related to the unambiguous discrimination of one GE line from all other lines, high-throughput sequencing represents, however, non-negligible help in solving this problem. In this context, precise and detailed sequence information on the genomic background is necessary to identify additional key targets, such as SNVs specifically associated with a particular genotype, cultivar, or variety. However, this approach requires access to appropriate databases with high-quality sequence data in order to compare a given sample to reference genome sequences, allowing us to identify specific single mutation points and genetic elements within a particular genetic background. Ideally, such database(s) will need to represent the entire diversity of all genotypes, cultivars, and varieties for each species of interest to aid the specific identification of a particular GE organism. Nonetheless, such crucial information is currently not available. Moreover, even if the specific GE line could be identified, other genetic modifications may be naturally introduced during vegetative propagation or during crossing with other cultivars. Therefore, it would be very difficult to trace genotypes associated with the offspring of GE plants if the sequences from these offspring are not included in reference databases [2,18,19,60]. This represents a major bottleneck for the detection of GE organisms by enforcement laboratories, making the data analysis even more complex. This problem is increased with GE plants carrying SNVs by reason of divergences in international legislation to determine if a GE organism falls under the scope of GMO legislation or not [2,61,62]. Consequently, the traceability and associated sequence information will be even more complex to obtain from GE organisms considered as GMOs in Europe but not in other countries. 

Although also subject to the aforementioned constraints regarding database reference information, shotgun metagenomics constitutes an interesting alternative to the targeted high-throughput sequencing approach because it requires no prior enrichment and it allows us to sequence an entire sample without any prior knowledge. However, no investigation has currently been performed to detect GM plants in the food/feed chain [63]. 

## 5. Conclusions

For the first time, the present proof-of-concept study has delivered experimental-based results on the performance of a targeted high-throughput sequencing approach to detect a specific SNV introduced into a CRISPR/Cas9 GE plant. These results highlight the potential performance compatibility of the proposed targeted high-throughput sequencing approach with the detection of GE plants in the GMO field. However, although this approach is highly promising, its current implementation in the context of GMO control remains challenging. Therefore, it will be essential for future case studies using this approach to generate sufficient data for further performance assessments. In addition, it is important to explore how this approach will be able to deal with major technical and analytical bottlenecks associated, for example, with the large size and complexity of plant genomes or with complex food/feed products. Moreover, if a GE organism is commercialized on the European market, a full validation process of the proposed sequencing approach, including transferability and robustness assessments, will be initially necessary before its use for control by enforcement laboratories.

## Figures and Tables

**Figure 1 foods-12-00455-f001:**
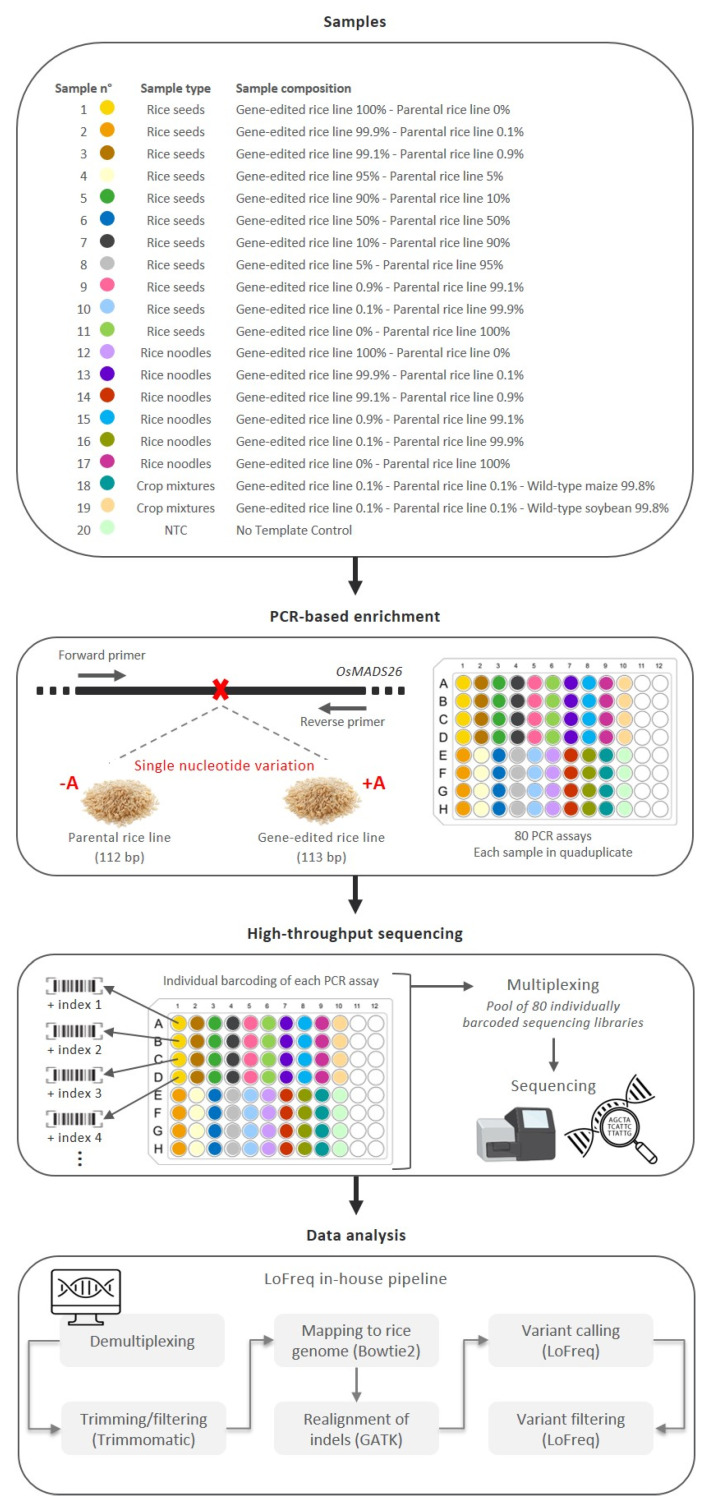
Schematic workflow of the targeted high-throughput sequencing approach performed in this study. A total of 20 different samples were analyzed using the proposed targeted high-throughput sequencing approach, including 19 samples containing various percentages of the GE rice line as well as 1 NTC. The proposed targeted high-throughput sequencing approach is composed of three successive steps. First, a PCR-based enrichment step is applied to the whole DNA extracted from a given sample. The used primer pair was previously designed to amplify the *OsMADS26* region carrying an SNV of interest (insertion or deletion of a single adenosine), resulting in a PCR amplicon of 113 bp for the GE rice line and a PCR amplicon of 112 bp for its parental rice line [16]. For each of these 20 samples, the PCR assay was performed in quadruplicate, resulting in a total of 80 final PCR products. Second, each PCR product is individually coupled to a unique barcode and then pooled for high-throughput sequencing. Third, raw sequencing reads are analyzed using an in-house pipeline based on LoFreq, developed to identify SNVs as well as to estimate their frequency.

**Table 1 foods-12-00455-t001:** Observed frequency of the CRISPR/Cas9 adenosine insertion, specific to the GE rice line in rice seed samples (n°1–11). The samples were composed of DNA extracted from rice seeds of the GE rice line and its parental rice line. The samples contained 100%, 99.9%, 99.1%, 95%, 90%, 50%, 10%, 5%, 0.9%, 0.1%, or 0% of the GE rice line. For each sample, the copy number associated with the GE rice line, previously measured using a 2-plex ddPCR method [16], is indicated (Appendix A). The targeted high-throughput sequencing approach was performed in quadruplicate for each sample, allowing us to subsequently determine the allele frequency of the CRISPR/Cas9 adenosine insertion specific to the GE rice line. The associated depth of coverage of the reference at the position of this specific indel (bp 679,646) is indicated.

Rice Seed Sample Description			CRISPR/Cas9 Adenosine Insertion				
Sample n°	GE Rice Line Content		Allele Frequency ~			Depth of Coverage ~	
	Percentage	Copy Number	Detection	Percentage	StDev	Value	StDev
**1**	100	13,733.3	+	99.92	0.03	119,592	29,499
**2**	99.9	13,346.7	+	99.86	0.03	97,038	25,007
**3**	99.1	13,273.3	+	99.73	0.02	70,873	12,339
**4**	95	12,113.3	+	97.97	0.15	101,581	63,687
**5**	90	10,753.3	+	92.06	0.24	75,787	36,739
**6**	50	5640.0	+	53.80	0.57	71,791	19,099
**7**	10	1153.3	+	8.01	0.29	86,841	10,531
**8**	5	528.0	+	4.00	0.15	98,283	40,588
**9**	0.9	85.3	+	0.66	0.01	101,464	33,345
**10**	0.1	10.8	+ *	0.13 *	0.02	126,724	42,631
**11**	0	0	−	0.00	0.00	158,436	99,002

~ For each sample, the average value of the 4 replicates and the associated standard deviation (StDev) are indicated. * The CRISPR/Cas9 adenosine insertion was observed in 3 of the 4 replicates.

**Table 2 foods-12-00455-t002:** Observed frequency of the CRISPR/Cas9 adenosine insertion, specific to the GE rice line in rice noodle samples (n°12–17). The samples were composed of DNA extracted from rice noodles of the GE rice line and its parental rice line. The samples contained 100%, 99.9%, 99.1%, 0.9%, 0.1%, or 0% of the GE rice line. For each sample, the copy number associated with the GE rice line, measured using a 2-plex ddPCR method [16], is indicated (Appendix A). The targeted high-throughput sequencing approach was performed in quadruplicate for each sample, allowing us to subsequently determine the allele frequency of the CRISPR/Cas9 adenosine insertion specific to the GE rice line. The associated depth of coverage of the reference at the position of this specific indel (bp 679,646) is indicated.

Rice noodle Sample Description			CRISPR/Cas9 Adenosine Insertion				
Sample n°	GE Rice Line Content		Allele Frequency ~			Depth of Coverage ~	
	Percentage	Copy Number	Detection	Percentage	StDev	Value	StDev
12	100	13,610.0	+	99.85	0.06	120,711	60,941
13	99.9	13,596.4	+	99.68	0.15	47,261	27,646
14	99.1	13,487.0	+	99.05	0.16	69,972	26,283
15	0.9	122.5	+	1.03	0.03	62,408	16,122
16	0.1	13.6	+	0.24	0.04	68,359	8826
17	0	0	−	0.00	0.00	67,906	37,437

~ For each sample, the average value of the 4 replicates and the associated standard deviation (StDev) are indicated.

**Table 3 foods-12-00455-t003:** Observed frequency of the CRISPR/Cas9 adenosine insertion, specific to the GE rice line in crop mixture samples (n°18–19). (A) The samples were composed of 0.1% of the GE rice line (corresponding to 14 estimated haploid genome copies), 0.1% of the parental rice line (corresponding to 14 estimated haploid genome copies), and 99.8% of WT maize (sample n°12) or WT soybean (sample n°13) (corresponding to 13,972 estimated haploid genome copies). (B) For each sample, the copy number associated with the GE rice line, previously measured using a 2-plex ddPCR method [16], is indicated (Appendix A). Additionally, regarding the rice ingredient, the rice ratio of the GE rice line is specified. The targeted high-throughput sequencing approach was performed in quadruplicate for each sample, allowing us to subsequently determine the allele frequency of the CRISPR/Cas9 adenosine insertion specific to the GE rice line. The associated depth of coverage of the reference at the position of this specific indel (bp 679,646) is indicated.

**(A)**							
**Sample Composition**							
**Sample n°18**	0.1 % GE rice line (14 estimated haploid genome copies)						
	0.1 % parental rice line (14 estimated haploid genome copies)						
	99.8 % WT maize (113,972 estimated haploid genome copies)						
**Sample n°19**	0.1 % GE rice line (14 estimated haploid genome copies)						
	0.1 % parental rice line (14 estimated haploid genome copies)						
	99.8 % WT soybean (113,972 estimated haploid genome copies)						
**(** **B** **)**							
**Crop Mixture Sample Description**			**CRISPR/Cas9 Adenosine Insertion**				
**Sample n°**	**GE Rice Line Content**		**Allele Frequency ~**			**Depth of Coverage ~**	
	**Copy Number**	**Rice Ratio ***	**Detection**	**Percentage**	**StDev**	**Value**	**StDev**
18	10.6	50	+	40.46	10.78	533	50
19	9.3	50	+	48.52	18.12	23,124	3181

~ For each sample, the average value of the 4 replicates and the associated standard deviation (StDev) are indicated. * The rice ratio of the GE rice line is determined according to the total amount of rice (ingredient) present in the investigated samples.

## Data Availability

The original contributions presented in the study are included in the article/Appendix A. Raw sequencing data are deposited in the European Nucleotide Archive under study accession number PRJEB58426. Further inquiries can be directed to the corresponding author.

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
