# Peer review of "Targeted High-Throughput Sequencing Enables the Detection of Single Nucleotide Variations in CRISPR/Cas9 Gene-Edited Organisms"

_foods, 2023, doi:10.3390/foods12030455_

Round 1
Reviewer 1 Report
It is my opinion that the authors demonstrated the efficiency of the proposed methodology with clarity and methodological robustness. The article seems well written and easy to understand.
This the research addressed that Gene-edited organisms and their derived food/feed products commercialized on market have to be controlled in the food/feed chain by GMO enforcement laboratories as required by the competent authorities to guarantee food/feed safety and traceability
Gene-edited organisms detection requires methodological approaches that can target and detect a specific single nucleotide variation introduced into a gene-edited organism. In this study, a workflow for targeted high-throughput sequencing was developed . Assessment of sensitivity and applicability of the devloped approach was also evaluated.
I believe that the topic is relevant in the field, because this approach highlights a robust and reliable detection methodology that could be adopted in the near future to detect specific mutational events resulting from plant editing, when a priori information about the modified gene variants has been provided by the developer of the edited plant. For example, to date, no plants with modified genome have been subject to the EU regulatory authorization procedure. Therefore, validated detection methods for genome-edited plants are not available to either the EU reference laboratory for GM food and feed or the national reference laboratories for GMO control in EU Member States. Similarly, the information needed to implement or develop a suitable detection method for product identification may not be readily available to EU authorities. Such an approach would be useful however, for all legislation imposing regulatory requirements for genome-edited plants
From my knowledge, the present study is the first available study using a targeted high-throughput sequencing approach to detect single nucleotide variations of interest in a gene-edited or ganism.
-The specific improvements should the authors consider is the methodology adopted is robustness enough
-In my opinion the references are appropriated
Author Response
It is my opinion that the authors demonstrated the efficiency of the proposed methodology with clarity and methodological robustness. The article seems well written and easy to understand. This the research addressed that Gene-edited organisms and their derived food/feed products commercialized on market have to be controlled in the food/feed chain by GMO enforcement laboratories as required by the competent authorities to guarantee food/feed safety and traceability. Gene-edited organisms detection requires methodological approaches that can target and detect a specific single nucleotide variation introduced into a gene-edited organism. In this study, a workflow for targeted high-throughput sequencing was developed. Assessment of sensitivity and applicability of the developed approach was also evaluated. I believe that the topic is relevant in the field, because this approach highlights a robust and reliable detection methodology that could be adopted in the near future to detect specific mutational events resulting from plant editing, when a priori information about the modified gene variants has been provided by the developer of the edited plant. For example, to date, no plants with modified genome have been subject to the EU regulatory authorization procedure. Therefore, validated detection methods for genome-edited plants are not available to either the EU reference laboratory for GM food and feed or the national reference laboratories for GMO control in EU Member States. Similarly, the information needed to implement or develop a suitable detection method for product identification may not be readily available to EU authorities. Such an approach would be useful however, for all legislation imposing regulatory requirements for genome-edited plants. From my knowledge, the present study is the first available study using a targeted high-throughput sequencing approach to detect single nucleotide variations of interest in a gene-edited organism.
Thank you.
-The specific improvements should the authors consider is the methodology adopted is robustness enough.
This point was added in the conclusion section : ”In case a GE organisms is commercialized on the European market, a full validation process of the proposed sequencing approach, including transferability and robustness assessment, will be initially required before its use for the control by enforcement laboratories.”
-In my opinion the references are appropriated.
Thank you.

Reviewer 2 Report
The manuscript explored the targeted high-throughput sequencing enables the detection of single nucleotide variations in CRISPR/Cas gene-edited organisms. They showed that a gene-edited rice line carrying a single adenosine insertion in OsMADS26 was performed to assess the feasibility and performance of the proposed targeted high-throughput sequencing approach. My overall evaluation of the manuscript is positive. There are a number of minor revisions, formal and scientific aspects that should be addressed.
1. Change CRISPR/Cas to CRISPR/Cas9 in the title and text section.
2. In the introduction section of line 102, only the European Union threshold is mentioned in GMO products, and it is necessary to mention this threshold for other countries, especially Japan.
3. The authors need to mention the off-target effect in the CRISPR/Cas9 system.
4. Mention the validation method to check the off-target effect.
5. Give a brief explanation about OsMADS26 in the introduction section.
6. Mention the concentration of the extracted DNA in the results section.
7. Mention the full term of ddPCR at the beginning of the text.
8. Considering that GMO products are not created based on a change in a single nucleotide, and mostly a genetic construction such as resistance to pests, resistance to glyphosate or increased expression of a gene in plants is designed. Why did they use this method? Are the usual methods not responsive? Mention this in the introduction.
Author Response
The manuscript explored the targeted high-throughput sequencing enables the detection of single nucleotide variations in CRISPR/Cas gene-edited organisms. They showed that a gene-edited rice line carrying a single adenosine insertion in OsMADS26 was performed to assess the feasibility and performance of the proposed targeted high-throughput sequencing approach. My overall evaluation of the manuscript is positive.
Thank you.
There are a number of minor revisions, formal and scientific aspects that should be addressed.
- Change CRISPR/Cas to CRISPR/Cas9 in the title and text section.
As requested, the term CRISPR/Cas was changed to CRISPR/Cas9 through the entire manuscript.
- In the introduction section of line 102, only the European Union threshold is mentioned in GMO products, and it is necessary to mention this threshold for other countries, especially Japan.
As suggested, a sentence specifying the threshold for GMO labelling in Japan was added.
- The authors need to mention the off-target effect in the CRISPR/Cas9 system.
This information was added in the introduction section.
- Mention the validation method to check the off-target effect.
The proposed targeted high-throughput sequencing approach is based on a prior PCR enrichment of +/- 100 bp. Consequently, even if off-targeting occurs, unless it happens in a region that would also be amplified by the primers, it would not interfere. In this study, no off-target mutations were observed in the sequences generated from the amplified regions of interest. This information is now indicated in the section 2.5.
At the genome level, possible off-target mutations could be investigated with whole-genome sequencing as recently suggested in few studies[1]. However, it still represents currently a challenging investigation, especially due to the large size and complexity of plant genomes as well as the availability of appropriate reference genomes. Further investigations to develop a mature analytical approach are therefore still necessary.
- Give a brief explanation about OsMADS26 in the introduction section.
As suggested, a brief explanation about OsMADS26 was added in the introduction section.
- Mention the concentration of the extracted DNA in the results section.
This information (DNA extraction yield: ~ 6x103 ng/g of rice) was added in the result section.
- Mention the full term of ddPCR at the beginning of the text.
As requested, the full term of ddPCR was specified at the beginning of the text (introduction section).
- Considering that GMO products are not created based on a change in a single nucleotide, and mostly a genetic construction such as resistance to pests, resistance to glyphosate or increased expression of a gene in plants is designed. Why did they use this method? Are the usual methods not responsive? Mention this in the introduction.
As suggested, in the introduction, few modifications were introduced to mention that both large modifications and single nucleotide variations are induced by gene-editing. We also mentioned that, in case of gene-edited organisms carrying a large nucleotide sequence variation, the development of detection methods is highly similar to GMO produced by classical genetic engineering.
[1] https://www.nature.com/articles/s41438-021-00549-4; https://www.ncbi.nlm.nih.gov/pmc/articles/PMC6587709/
